# GARCH Option Pricing Models and the Variance Risk Premium

**Wenjun Zhang [1],* and Jin E. Zhang [2]**

[1] Department of Mathematical Sciences, School of Engineering, Computer and Mathematical Sciences, Auckland University of Technology, Auckland 1142, New Zealand

[2] Department of Accountancy and Finance, Otago Business School, University of Otago, Dunedin 9054, New Zealand; jin.zhang@otago.ac.nz

* Correspondence: wenjun.zhang@aut.ac.nz

**Abstract:** In this paper, we modify Duan's (1995) local risk-neutral valuation relationship (mLRNVR) for the GARCH option-pricing models. In our mLRNVR, the conditional variances under two measures are designed to be different and the variance process is more persistent in the risk-neutral measure than in the physical one, so that one is able to capture the variance risk premium. Empirical estimation exercises show that the GARCH option-pricing models under our mLRNVR are able to price the SPX one-month variance swap rate, i.e., the CBOE Volatility Index (VIX) accurately. Our research suggests that one should use our mLRNVR when pricing options with GARCH models.

**Keywords:** GARCH option-pricing models; stochastic volatility; the CBOE VIX; variance risk premium

**JEL Classification:** G13; C52

## 1. Introduction

In this paper, we modify the local risk-neutral valuation relationship (mLRNVR) in the GARCH option-pricing models. The GARCH option-pricing model was first introduced by Duan (1995) with a locally risk-neutral valuation relationship (LRNVR), in which the conditional variances and model parameters remained the same under the physical measure and the risk-neutral measure. Since then, Duan's LRNVR has been widely used by finance researchers and practitioners in pricing options under the GARCH framework. However, as Barone-Adesi et al. (2008) pointed out, empirical evidence showed that the restriction in Duan's LRNVR led to rather poor pricing and hedging performances. Hao and Zhang (2013) showed explicitly that the GARCH option-pricing models under Duan's LRNVR under-priced S&P 500 (SPX) one-month variance swap rate, i.e., the Chicago Board Options Exchange (CBOE) Volatility Index (VIX) by about 10%. We propose a new mLRNVR in GARCH option models in order to resolve the issue of under-pricing by the LRNVR.

Since the seminal work of Bollerslev (1986); Engle (1982), the family of GARCH volatility models has been widely used in empirical asset pricing and financial risk management partly because of the likelihood function of asset returns in the GARCH models could often be expressed in a closed-form in terms of observed data. One can estimate the model parameters by using the maximum likelihood estimation (MLE) method, which is often a challenging task for most of the stochastic volatility models. Motivated by the success of GARCH models in fitting asset returns, Duan (1995) pioneered in applying GARCH models for the SPX option pricing by proposing the LRNVR. Duan's LRNVR has been followed by many papers. Ritchken and Trevor (1999) developed an efficient lattice algorithm to price European and American options under the GARCH processes. Chernov and Ghysels (2000) estimated physical and risk-neutral probabilities by using Heston (1993) and GARCH option models.

Heston and Nandi (2000) developed a closed-form option valuation formula for a spot asset whose variance follows a GARCH$(p, q)$ process that can be correlated with the returns of the spot asset. Christoffersen and Jacobs (2004) compared a variety of GARCH option-pricing models using option prices and asset returns. Christoffersen et al. (2008) extended the Heston-Nandi model to incorporate long-run and short-run volatility components for the valuation of European options. An interesting survey on the theoretical results and empirical evidence of GARCH option-pricing could be found in Christoffersen et al. (2013b). The evaluation of European options in the GARCH models with Levy jumps was discussed in Ornthanalai (2014). Kanniainen et al. (2014) evaluated SPX options using three variant GARCH models with VIX data. Their empirical evidence showed that a joint maximum likelihood estimation using SPX returns and VIX improved the performance of pricing SPX options compared with traditional MLE using the returns data only. Wang et al. (2017) priced the VIX futures with the Heston-Nandi GARCH model. All of these papers used Duan's LRNVR in changing probability measures from a physical one to a risk-neutral one. However, there exists a problem in pricing options under the GARCH models with the LRNVR.

Empirical evidence by Chernov and Ghysels (2000); Christoffersen and Jacobs (2004) showed that the GARCH option-pricing model under the LRNVR had poor pricing and hedging performances, but they did not realize that the problem came from the LRNVR. Barone-Adesi et al. (2008) were the first to explicitly point out that the poor performance came from the restriction required by the LRNVR. They proposed a new method for pricing options based on GARCH models with filtered historical innovations, in which they did not specify directly the change of measures, and proposed approximating them by calibrating a new set of risk-neutral GARCH parameters directly on market option prices. Christoffersen et al. (2013a) developed a GARCH option model with a new pricing kernel allowing for a variance premium. They presented an explicit relationship between physical and risk-neutral conditional variances and GARCH parameters, in which risk aversion and the market price of variance risk entered into their analytical formulas. It is interesting to note that the persistence parameter remained unchanged in their change of probability measures. Hao and Zhang (2013) developed a model for the CBOE VIX by using a GARCH option-pricing model under the LRNVR. They found that the model-implied VIX was about 10% lower than the market VIX due to the lack of a variance risk premium. Given that the VIX is a proxy of the SPX one-month variance swap rate, one may conclude that GARCH option-pricing models under LRNVR under-price variance swap and hence at-the-money options by about 10%. Despite these developments of literature pointing out the problem of LRNVR, later studies, e.g., Ornthanalai (2014); Kanniainen et al. (2014); Wang et al. (2017), still used it in pricing SPX options and other derivatives. They might have realized the problem with the LRNVR, but lacked a better one to resolve the issue. Developing a new risk-neutral valuation relationship in GARCH option-pricing models becomes an urgent task. That is the focus in this paper. Parallel to our work, Huang et al. (2017 2019) used a stochastic discount factor for risk-neutralization for option pricing and VIX futures pricing, respectively. The stochastic discount factor risk-neutralization is a generalization of the LRNVR. In contrast to our modification, the persistent parameter in the models remained unchanged under the stochastic discount factor risk-neutralization.

In this paper, we propose a modified local risk-neutral valuation relationship, referred to as the mLRNVR. In our mLRNVR, conditional variances under two measures are designed to be different and the GARCH process is more persistent in the risk-neutral measure than in the physical one, so that we are able to capture the negative variance risk premium. We display the improvement of the mLRNVR compared with the LRNVR using theoretical and empirical evidences. Specifically, we find the theoretical VIX squared value as the conditional risk-neutral expectation of the arithmetic mean variance over the next 21 trading days under the mLRNVR. The GARCH implied VIX formulas are derived using the features of square-root stochastic autoregressive volatility (SR-SARV) models. We apply several calibration methods to estimate the model parameters using various sets of time series data, and compare the theoretical formula performances with the market data. Various combinations of time series of the daily closing price of the S&P 500 index and the CBOE VIX are used to find the

maximum likelihood estimation of the GARCH models. The corresponding implied VIX time series are then calculated from the calibrated model. Similar to the empirical evidence in Hao and Zhang (2013) and Wang et al. (2017), when only S&P 500 returns are used for estimation, the GARCH implied VIX is consistently and significantly lower than the CBOE VIX. When the CBOE VIX is used for estimation, the implied VIX fits the statistical properties of the CBOE VIX and matches the CBOE VIX data better. The numerical results provide evidence that the GARCH option-pricing under the mLRNVR is more suitable to price variance swap. In the case of GARCH(1,1), we compare the diffusion limit of the GARCH process under the physical measure and the mLRNVR risk-neutral measure to show that the variance premium is captured in the risk-neutral dynamics.

This paper makes at least two contributions. First, by using a newly proposed mLRNVR, we have resolved the issue of under-pricing in GARCH option-pricing models under the LRNVR. Our research suggests that one should use our mLRNVR when pricing options with GARCH models. Second, we develop a model for the VIX in a GARCH framework by using the mLRNVR. Our model is able to capture negative variance risk premiums, and hence should be used in pricing VIX derivatives, including VIX futures and VIX options.

The article is structured as follows. Section 2 proposes a new risk-neutral valuation relationship for the GARCH(1,1). In Section 3, we derive a theoretical VIX formula for the GARCH(1,1) model under the mLRNVR, and extend the derivation idea to a broad class of GARCH models, which include GARCH, TGARCH, AGARCH and EGARCH models. In Section 4, we calibrate these GARCH models using various combinations of time series of the S&P 500 index and the CBOE VIX. Section 5 compares the CBOE VIX with the GARCH implied VIX obtained from the calibrated GARCH models. In Section 6, we analyze the diffusion limit of the GARCH process under the risk-neutral measure in mLRNVR to demonstrate that the risk-neutral dynamics captures the variance risk premium. We then conclude the findings in Section 7.

## 2. GARCH Model Specification

In this paper, we consider the asset price as a discrete-time stochastic process and denote the asset price at time $t$ as $X_t$. It was proposed in Duan (1995) that the returns of the asset follows a conditional lognormal distribution under the physical measure $P$ as

$$\ln \frac{X_t}{X_{t-1}} = r - \frac{1}{2}h_t + \lambda_1 \sqrt{h_t} + \epsilon_t, \tag{1}$$

where $r$ is the one-period risk-free interest rate, $\lambda_1$ is the asset risk premium, and $\epsilon_t$ follows a GARCH$(p,q)$ process introduced in Bollerslev (1986) with mean zero and conditional variance $h_t$

$$\epsilon_t | \phi_{t-1} \sim N(0, h_t) \quad \text{under measure } P,$$

$$h_t = \alpha_0 + \sum_{i=1}^{q} \alpha_i \epsilon_{t-i}^2 + \sum_{j=1}^{p} \beta_j h_{t-j}, \tag{2}$$

where $\phi_t$ is the information set of up to and including time $t$; $\alpha_0 \geq 0$, $\alpha_i \geq 0$ for $i = 1, 2, \ldots, q$ and $\beta_j \geq 0$ for $j = 1, 2, \ldots, p$. We focus on the GARCH(1,1) case, so the Equation (2) simplifies to

$$h_t = \alpha_0 + \alpha_1 \epsilon_{t-1}^2 + \beta_1 h_{t-1}. \tag{3}$$

In order to accommodate the heteroskedasticity of the asset returns process in (1), Duan (1995) introduced the LRNVR under which the expected return should be the risk-free rate and the

one-step-ahead variance should be the same in both measures. Specifically, under the LRNVR, the dynamics of asset returns in the risk-neutral pricing measure $Q$ has the form

$$\ln \frac{X_t}{X_{t-1}} = r - \frac{1}{2}h_t + \xi_t, \quad \xi_t|\phi_{t-1} \sim N(0, h_t) \quad \text{under measure } Q,$$

$$h_t = \alpha_0 + \alpha_1 \left( \xi_{t-1} - \lambda_1 \sqrt{h_{t-1}} \right)^2 + \beta_1 h_{t-1}. \tag{4}$$

It is well documented in Bollerslev et al. (2009); Carr and Wu (2009) that variance risk premiums come from either its correlation with the return risk and return risk premium, or a separate premium of the independent variance variation. It is also established in Carr and Wu (2009) that the majority of the market variance risk premium is generated by an independent variance risk factor. Motivated by the empirical findings that the market variance risk premiums is closely related to the variance variation, we propose an alternative risk-neutral valuation relationship to the LRNVR which can capture the variance risk premium as follows.

**Definition 1.** *A pricing measure Q is said to satisfy the modified local risk-neutral valuation relationship (mLRNVR) if the dynamics of asset returns in the risk-neutral pricing measure Q under the mLRNVR has the following form*

$$\ln \frac{X_t}{X_{t-1}} = r - \frac{1}{2}h_t + \xi_t, \quad \xi_t|\phi_{t-1} \sim N(0, h_t) \quad \text{under measure } Q,$$

$$h_t = \alpha_0 + \alpha_1 \left( \xi_{t-1} - \lambda_1 \sqrt{h_{t-1}} \right)^2 + \beta_1^* h_{t-1}$$

$$= \alpha_0 + \alpha_1 \left( \xi_{t-1} - \lambda_1 \sqrt{h_{t-1}} \right)^2 + (\beta_1 - \sqrt{2}\alpha_1\lambda_2)h_{t-1}. \tag{5}$$

**Remark 1.** *In the classic continuous-time stochastic volatility models, e.g., the Heston model (Heston 1993), the instantaneous variances are the same under the physical and risk-neutral measures. However, in the discrete-time GARCH models, the conditional variances are defined in a finite-time interval which is usually one day in practice. The conditional variances should be designed to be different under the physical and risk-neutral measures, as in the mLRNVR, to incorporate the variance risk premium during the finite-time interval.*

The proposed GARCH(1,1) process (5) is different to the one derived by Duan (1995) under the LRNVR. Under the mLRNVR the persistence parameter is designed to be different in the $P$ and $Q$ measures, whereas under the LRNVR the persistence parameter is the same in the $P$ and $Q$ measures. Specifically, for the dynamics of risk-neutral measure $Q$ under the mLRNVR, the persistence parameter of conditional variance is $\beta_1^* = \beta_1 - \sqrt{2}\alpha_1\lambda_2$, where $\lambda_2$ represents the variance risk premium of the asset. Note that there is a negative sign in front of the variance risk premium $\lambda_2$ which is estimated to be negative from empirical studies Bollerslev et al. (2009); Carr and Wu (2009). Hence, we expect the variance process is more persistent in the risk-neutral measure than that in the physical one. The motivation for the inclusion of the variance risk premium is discussed in Hao and Zhang (2013), where it was shown that there is no risk adjustment for the variance risk of the process in Duan (1995) from the physical measure to the risk-neutral measure under the LRNVR. It was also discussed in Barone-Adesi et al. (2008) and Christoffersen et al. (2013a) that the restriction of conditional volatility of historical and risk-neutral pricing distributions with the same model parameters leads to poor calibration results in the empirical studies (cf. Chernov and Ghysels 2000; Christoffersen et al. 2006; Hao and Zhang 2013). Therefore, it was suggested that the parameters of volatility dynamics of historical and risk-neutral pricing returns might be different in Barone-Adesi et al. (2008). We adapt

the idea by modifying the persistence parameter in $Q$ to incorporate the variance risk premium in the model.[1] The theoretical justification of the modification is further discussed in Section 6.

## 3. VIX Formulas of the GARCH Models

The CBOE introduced the VIX index in 1993. The VIX was calculated from the implied volatilities of the eight near-the-money, nearby, and second nearby S&P 100 index options based on the methodology of Whaley (Whaley 1993). The VIX was a proxy of the implied volatility of 30 calendar days at-the-money (ATM) options. In 2003, the CBOE used another theory proposed in Carr and Madan (1998) and Demeterfi et al. (1999) to design a new methodology to compute the CBOE's VIX. The new VIX is based on the prices of a portfolio of 30 calendar days out-of-the-money (OTM) S&P 500 index call and put options. The square of the new VIX represents the S&P 500 30-day variance swap rate. The old VIX has been renamed the VXO. We use $\text{VIX}^{\text{Mkt}}$ to denote the CBOE VIX which is computed using market option prices.

The CBOE VIX index reflects investors' expectation of the volatility of the S&P 500 in the next 30 calendar days or 21 trading days. Following Hao and Zhang (2013), we have calculated the VIX index implied by the GARCH models as the mean value of the expected variance in the $n$ sub-periods of the next 21 trading days, that is

$$\left( \frac{\text{VIX}_t^{\text{Imp}}}{100} \right)^2 = \frac{1}{n} \sum_{k=1}^{n} E_t^Q \left( h_{t + \frac{\tau_0 k}{n}} \right), \tag{6}$$

where $\tau_0$ represents 21 trading days and $\text{VIX}_t^{\text{Imp}}$ stands for the model implied VIX index. In particular, we use the daily closing value data, so it implies $\tau_0 = n$, and

$$V_t = \frac{1}{n} \sum_{k=1}^{n} E_t^Q \left( h_{t+k} \right), \tag{7}$$

where the term $V_t = \frac{1}{252} \left( \frac{\text{VIX}_t^{\text{Imp}}}{100} \right)^2$ is defined as a function of $\text{VIX}_t^{\text{Imp}}$ to measure the expected daily variance of the S&P 500. The conditional mean of the future variance can be calculated in a broad class of GARCH models, as discussed in Hao and Zhang (2013) and Wang et al. (2017).

We derive the implied VIX from the model (5) under $Q$ by first rewriting the error terms of the process using the standard normal distribution as

$$\begin{aligned} \ln \frac{X_t}{X_{t-1}} &= r - \frac{1}{2} h_t + \sqrt{h_t} \epsilon_t, \\ h_t &= \alpha_0 + \alpha_1 h_{t-1} \left( \epsilon_{t-1} - \lambda_1 \right)^2 + (\beta_1 - \sqrt{2} \alpha_1 \lambda_2) h_{t-1}, \end{aligned} \tag{8}$$

where $\epsilon_t$ is the standard normal random variable, conditional on the information set up to and including time $t - 1$ under $Q$.

---

[1] Our purpose is trying to propose a simple modification to the LRNVR in order to incorporate the variance risk premium under the risk-neutral measure in the GARCH models. It is noted that in Huang et al. (2019), with an additional source of uncertainty in the conditional variance under the physical measure, the Radon-Nikodym derivative was derived and the variance risk premium was incorporated in the risk-neutral measure. However, these two approaches differ, since our mLRNVR method only changes the persistence parameter without introducing an extra uncertainty.

One can rewrite the GARCH$(1,1)$ process (8) as a special case of the square-root stochastic autoregressive volatility (SR-SARV(1)) models introduced in Meddahi and Renault (2004) with the following form

$$h_{t+1} = \omega + \gamma h_t + v_t, \text{ with } \mathbb{E}\left[v_t | \phi_{t-1}\right] = 0,$$
$$\omega = \alpha_0, \quad \gamma = \alpha_1(1 + \lambda_1^2) + \beta_1 - \sqrt{2}\alpha_1\lambda_2, \tag{9}$$
$$v_t = \alpha_1 h_t(\epsilon_t^2 - 1 - 2\lambda_1\epsilon_t).$$

It was shown in Hao and Zhang (2013) that if the S&P 500 returns follow a SR-SARV($p$) process under the risk-neutral measure, then the implied daily variance $V_t$ at time $t$ is affine in the conditional variance $h_{t+1}$. Following similar ideas, we can obtain the long term variance as $\bar{h} = \lim_{m \to \infty} \mathbb{E}_t^Q[h_{t+m}] = \frac{\omega}{1-\gamma}$ by noticing

$$\bar{h} = \lim_{m \to \infty} \mathbb{E}_t^Q[h_{t+m}] = \lim_{m \to \infty} \mathbb{E}_t^Q[\omega + \gamma h_{t+m-1} + v_{t+m-1}] = \omega + \gamma \bar{h}. \tag{10}$$

Then the conditional expectation of the variance after two periods can be obtained via the long-run variance:

$$\mathbb{E}_t^Q\left[h_{t+2}\right] - \bar{h} = \mathbb{E}_t^Q\left[\omega + \gamma h_{t+1} + v_{t+1}\right] - \frac{\omega}{1-\gamma} = \omega + \gamma h_{t+1} - \frac{\omega}{1-\gamma} = \gamma(h_{t+1} - \bar{h}). \tag{11}$$

So the conditional expectation of the variance after $n$ periods is given by

$$\mathbb{E}_t^Q\left[h_{t+n}\right] = \bar{h} + \gamma^{n-1}(h_{t+1} - \bar{h}). \tag{12}$$

Therefore, we can represent the expected daily variance as an affine function of $h_{t+1}$

$$\begin{aligned}
V_t &= \frac{1}{n}\sum_{k=1}^{n} \mathbb{E}_t^Q\left(h_{t+k}\right) \\
&= \bar{h} + \frac{1}{n}\sum_{k=1}^{n} \gamma^{k-1}(h_{t+1} - \bar{h}) \\
&= \bar{h} + \frac{1-\gamma^n}{n(1-\gamma)}(h_{t+1} - \bar{h}) \\
&= \left(1 - \frac{1-\gamma^n}{n(1-\gamma)}\right)\frac{\omega}{1-\gamma} + \frac{1-\gamma^n}{n(1-\gamma)}h_{t+1} \\
&= A + Bh_{t+1},
\end{aligned} \tag{13}$$

where $A = \frac{(1-B)\omega}{1-\gamma}$ and $B = \frac{1-\gamma^n}{n(1-\gamma)}$.

Apart from the GARCH$(1,1)$ model discussed above, we also consider the threshold GARCH$(1,1)$ (TGARCH) model introduced in Glosten et al. (1993), the non-linear asymmetric GARCH$(1,1)$ (AGARCH) model proposed in Engle and Ng (1993) and the exponential GARCH$(1,1)$ (EGARCH) model by Nelson (Nelson 1991). The forms of the models in the physical measure $P$ and in the risk-neutral measure $Q$ under the mLRNVR are as follows:

TGARCH$(1,1)$

Physical measure: $h_t = \alpha_0 + \alpha_1\epsilon_{t-1}^2 + \theta\epsilon_{t-1}^2 \mathbf{1}(\epsilon_{t-1} < 0) + \beta_1 h_{t-1}$, \hfill (14)

mLRNVR: $h_t = \alpha_0 + \left(\xi_{t-1} - \lambda_1\sqrt{h_{t-1}}\right)^2 \left(\alpha_1 + \theta\mathbf{1}(\xi_{t-1} - \lambda_1\sqrt{h_{t-1}} < 0)\right) + (\beta_1 - \sqrt{2}\alpha_1\lambda_2)h_{t-1}$.

AGARCH(1,1)

$$\text{Physical measure: } h_t = \alpha_0 + \alpha_1 \left( \epsilon_{t-1} - \theta \sqrt{h_{t-1}} \right)^2 + \beta_1 h_{t-1}, \tag{15}$$

$$\text{mLRNVR: } h_t = \alpha_0 + \alpha_1 \left( \xi_{t-1} - \lambda_1 \sqrt{h_{t-1}} - \theta \sqrt{h_{t-1}} \right)^2 + (\beta_1 - \sqrt{2}\alpha_1\lambda_2) h_{t-1}.$$

EGARCH(1,1)

Physical measure:

$$\ln h_t = \alpha_0 + \beta_1 \ln h_{t-1} + \alpha_1 \frac{\epsilon_{t-1}}{\sqrt{h_{t-1}}} + \kappa \left( \left| \frac{\epsilon_{t-1}}{\sqrt{h_{t-1}}} \right| - \sqrt{\frac{2}{\pi}} \right), \tag{16}$$

mLRNVR:

$$\ln h_t = \alpha_0 + (\beta_1 - \sqrt{2}\alpha_1\lambda_2) \ln h_{t-1} + \alpha_1 \left( \frac{\epsilon_{t-1}}{\sqrt{h_{t-1}}} - \lambda_1 \right) + \kappa \left( \left| \frac{\epsilon_{t-1}}{\sqrt{h_{t-1}}} - \lambda_1 \right| - \sqrt{\frac{2}{\pi}} \right).$$

As shown in Hao and Zhang (2013), these widely used GARCH models are special cases of SR-SARV($p$) models, and following similar derivation process as the GARCH(1,1) model, we can obtain the implied VIX formula for different GARCH models analogous to the ones obtained in Hao and Zhang (2013). For the convenience of readers, we list the implied VIX formula as follows:

TGARCH(1,1)

$$V_t = C + Dh_{t+1}, \tag{17}$$

where

$$
\begin{aligned}
C &= \frac{\alpha_0(1-D)}{1-\eta}, \\
D &= \frac{1-\eta^n}{n(1-\eta)}, \\
\eta &= \alpha_1(1+\lambda_1^2) + (\beta_1 - \sqrt{2}\alpha_1\lambda_2) + \theta S, \\
S &= \frac{\lambda_1}{\sqrt{2\pi}} e^{-\frac{\lambda_1^2}{2}} + (1+\lambda_1^2)N(\lambda_1).
\end{aligned}
$$

Note that $N(\cdot)$ denotes the cumulative function of the normal distribution.

AGARCH(1,1)

$$V_t = E + Fh_{t+1}, \tag{18}$$

where

$$
\begin{aligned}
E &= \frac{\alpha_0(1-F)}{1-\eta}, \\
F &= \frac{1-\eta^n}{n(1-\eta)}, \\
\eta &= \alpha_1(1+(\lambda_1+\theta)^2) + (\beta_1 - \sqrt{2}\alpha_1\lambda_2).
\end{aligned}
$$

EGARCH(1,1)

$$V_t = \frac{1}{n} \left( h_{t+1} + \sum_{k=1}^{n-1} \left( \prod_{i=0}^{k-1} l_i \right) h_{t+1}^{(\beta_1 - \sqrt{2}\alpha_1\lambda_2)^k} \right), \tag{19}$$

where

$$
\begin{aligned}
l_i = e^{(\beta_1 - \sqrt{2}\alpha_1\lambda_2)^i\left(\alpha_0 - \kappa\sqrt{\frac{2}{\pi}}\right)} \\
\left(e^{-(\beta_1 - \sqrt{2}\alpha_1\lambda_2)^i(\alpha_1-\kappa)\lambda_1 + 0.5(\beta_1 - \sqrt{2}\alpha_1\lambda_2)^{2i}(\alpha_1-\kappa)^2}N(\lambda_1 - (\beta_1 - \sqrt{2}\alpha_1\lambda_2)^i(\alpha_1 - \kappa)) \right. \\
\left. + e^{-(\beta_1 - \sqrt{2}\alpha_1\lambda_2)^i(\alpha_1+\kappa)\lambda_1 + 0.5(\beta_1 - \sqrt{2}\alpha_1\lambda_2)^{2i}(\alpha_1+\kappa)^2}N((\beta_1 - \sqrt{2}\alpha_1\lambda_2)^i(\alpha_1 + \kappa) - \lambda_1)\right)
\end{aligned}
$$

## 4. Data and Estimation

It was shown in Hao and Zhang (2013) that under the LRNVR, the GARCH implied VIX does not fit the market data of the CBOE VIX very well. The model was analyzed to show that the reason may be that the variance risk premium and the volatility risk price were not present in the diffusion limit of the GARCH models under the LRNVR. In the modified GARCH processes, we include the variance risk premium in the models under the mLRNVR. It is also of interest to see whether the implied VIX in the modified GARCH models fit the CBOE VIX market values better. In this section, we will investigate this question by estimating the parameters in the modified GARCH models and calculating the corresponding GARCH implied VIX time series for comparison with the CBOE VIX.

The time-series data we use for the GARCH models calibration are the closing values of the S&P 500 and the CBOE VIX ranging from 2nd January 1990 to 30th June 2017. For the daily risk-free interest rate, we use the three-month Treasury bill secondary market rate from the U.S. Federal Reserve website.

There are different methods to calibrate the models using market data. We will use the common maximum likelihood approach to estimate the parameters of the models. We can use only the S&P 500 returns data to obtain a maximum likelihood estimation of the GARCH processes under the physical measure $P$ and fix the variance risk premium parameter $\lambda_2 = 0$, since $\lambda_2$ is not included in the GARCH models under the $P$ measure. For the S&P 500 returns data only, the log-likelihood function $\ln L_R$ for the GARCH models is given by

$$
\ln L_R = -\frac{T\ln(2\pi)}{2} - \frac{1}{2}\sum_{t=1}^{T}\left(\ln(h_t) + \left(\ln\frac{X_t}{X_{t-1}} - r - \lambda_1\sqrt{h_t} + \frac{h_t}{2}\right)^2 / h_t\right), \tag{20}
$$

where the conditional variance $h_t$ is updated by corresponding processes using different forms of GARCH models and $X_t$ is defined in equation (1) as the asset price. For the maximum likelihood estimation, the conditional variance for the first period is set as the variance of S&P 500 returns over the whole sample period. The stationary conditions for the GARCH processes under the physical and the risk-neutral measures are different, with the latter having stricter constraints on the parameters. Thus, we find the estimation of the parameters in the GARCH models by maximizing the corresponding log-likelihood function subject to the stationary conditions under the risk-neutral measures.

We may also calibrate the GARCH models by matching the model implied VIX to the market value of the CBOE VIX, since the CBOE VIX series may contain additional information about the underlying S&P 500 returns process. To utilize both time series, we follow the assumption in Hao and Zhang (2013) that the pricing differences between the CBOE VIX and the implied VIX on a daily basis come from a normal distribution

$$
\text{VIX}^{\text{Mkt}} = \text{VIX}^{\text{Imp}} + \mu, \quad \mu \sim N(0, s^2), \tag{21}
$$

where $s^2$ is estimated using the sample variance of pricing difference $\hat{s}^2 = \text{var}(\text{VIX}^{\text{Mkt}} - \text{VIX}^{\text{Imp}})$. Under the above assumption, the log-likelihood function corresponding to the CBOE VIX data is

$$
\ln L_V = -\frac{T\ln(2\pi\hat{s}^2)}{2} - \sum_{t=1}^{T}\frac{\left(\text{VIX}^{\text{Mkt}} - \text{VIX}^{\text{Imp}}\right)^2}{2\hat{s}^2}. \tag{22}
$$

Alternative to the method discussed in Hao and Zhang (2013), it was suggested in Kanniainen et al. (2014) to use the following model to describe autoregressive disturbances:

$$\mu_t = \rho\mu_{t-1} + e_t, \tag{23}$$

where $\mu_t = \text{VIX}_t^{\text{Mkt}} - \text{VIX}_t^{\text{Imp}}$ and $e_t \sim \text{NID}(0, \sigma^2)$. Based on the autoregressive disturbance process, we can estimate the variance of pricing difference $\hat{s}^2$ in terms of the autoregressive disturbance correlation $\rho$ and the variance $\hat{\sigma}^2$ obtained from the sample. The formula can be specified as $\hat{s}^2 = \frac{\hat{\sigma}^2}{1-\rho^2}$ from the relation

$$\text{Var}(\mu_t) = \text{Var}(\rho\mu_{t-1} + e_t) \implies s^2 = \rho^2 s^2 + \sigma^2. \tag{24}$$

The log-likelihood function corresponding to the CBOE VIX data is

$$\ln L_V = -\frac{T \ln(2\pi \frac{\hat{\sigma}^2}{1-\rho^2})}{2} - \sum_{t=1}^{T} \frac{\left(\text{VIX}^{\text{Mkt}} - \text{VIX}^{\text{Imp}}\right)^2}{2\frac{\hat{\sigma}^2}{1-\rho^2}}. \tag{25}$$

We will compare the performance of this autoregressive disturbance process to the mLRNVR process in the numerical simulation section.

Apart from using the S&P 500 returns data and CBOE VIX data for calibration of the GARCH models separately, we also combine both time series to find a joint maximum likelihood estimation of the models by maximizing the joint log-likelihood function

$$\ln L_T = \ln L_R + \ln L_V. \tag{26}$$

## 5. Numerical Results

In this section, we compare the estimated parameters from different data used for calibration.[2] In particular, the output tables display the maximum likelihood estimates and the standard errors of the GARCH models. The values of the log-likelihood functions (20), (22), (25), (26) are also displayed in the tables. Although the contributions from the S&P 500 returns and the CBOE VIX as well as the joint likelihood values are reported, we maximize the function $\ln L_R$ when only S&P 500 returns are used, the function $\ln L_V$ when only CBOE VIX data are used and the function $\ln L_T$ when both time series are used. All the output tables show the comparison results among the LRNVR process, the mLRNVR process and the LRNVR process with an autoregressive coefficient (referred to as the generalized LRNVR process in the following). We can observe that the mLRNVR process has the largest maximum likelihood values compared with the LRNVR and the generalized LRNVR processes in all GARCH models. The maximum likelihood values of the generalized LRNVR processes are generally better than those of the LRNVR processes. It is as expected since the generalized LRNVR process has an extra autoregressive coefficient $\rho$ compared with the LRNVR process. However, the extra autoregressive coefficient $\rho$ is not statistically significant in all GARCH models. So the generalized LRNVR process is not significantly better than the LRNVR process, and we will not discuss the numerical results of the generalized LRNVR process in details and focus on the comparison between the mLRNVR and LRNVR processes.

From the output in Table 1, we can see that the equity risk premium $\lambda_1$ increases significantly from 0.0886 (returns data used) to 0.2134 (both data used) and 0.2253 (VIX data used) in the GARCH(1,1) models when the CBOE VIX data are used for calibration. The variance risk premium $\lambda_2$ is negative and significantly different from zero as $-0.3670$ (both sets of data used) and $-0.3514$ (VIX data used). The persistence of the conditional variance $\beta_1$ increases slightly from 0.8543 (returns data used) to

---

[2]    We have also tried the out-of-sample prediction. Note that the relation between $\text{VIX}_t^2$ and the latent process $h_t$ is linear according to Equation (13). Predicting $\text{VIX}_t^2$ is essentially forecasting the latent process $h_t$. This task seems to be not straightforward in the GARCH models, hence is left for further research.

0.9251 (both sets of data used) and 0.9286 (VIX data used). There is a sizable decrease of the parameter value $\alpha_1$ from 0.1256 (returns data used) to 0.0474 (both sets of data used) and 0.0456 (VIX data used). Comparing the maximum likelihood result of the model under the mLRNVR and the results under the LRNVR, we can see that the maximum likelihood value increases significantly from 54,697 to 55,921 (both sets of data used) and from 33,424 to 33,662 (VIX data used).

Similar numerical results are also observed in the other types of GARCH models as displayed in Tables 2–4. Specifically, Table 2 shows that the equity risk premium $\lambda_1$ increases significantly from 0.0131 (returns data used) to 0.1160 (both sets of data used) and 0.0889 (VIX data used) in the TGARCH(1,1) model when the CBOE VIX data are used for calibration. The variance risk premium $\lambda_2$ is negative and significantly different from zero as −0.4112 (both sets of data used) and −0.3978 (VIX data used). The persistence of conditional variance, $\beta_1$ increases significantly from 0.8338 (returns data used) to 0.9561 (both sets of data used) and 0.9553 (VIX data used). There is a decrease of the parameter value $\alpha_1$ from 0.0256 (returns data used) to 0.0091 (both sets of data used) and 0.0060 (VIX data used). Comparing the maximum likelihood result of the TGARCH(1,1) model under the mLRNVR and the results under the LRNVR, we can see that the maximum likelihood value increases significantly from 55,455 to 56,282 (both sets of data used) and from 33,468 to 33,795 (VIX data used).

Table 3 shows the calibration results of the AGARCH(1,1) model using both returns and VIX data. If using VIX data only, it is not easy to distinguish the parameters $\theta$ and $\lambda_1$. Therefore, the numerical results using VIX data are not displayed in the table. From Table 3 we observe that the equity risk premium $\lambda_1$ increases significantly from 0.0255 (returns data used) to 0.1158 (both sets of data used) in the AGARCH(1,1) model when the CBOE VIX data and returns are used for calibration. The variance risk premium $\lambda_2$ is negative and significantly different from zero as −0.3125 (both sets of data used). The persistence of conditional variance $\beta_1$ increases from 0.8810 (returns data used) to 0.9316 (both sets of data used). There is a big decrease in the parameter value $\alpha_1$ from 0.0841 (returns data used) to 0.0380 (both sets of data used). Comparing the maximum likelihood result of the AGARCH(1,1) model under the mLRNVR and the results under the LRNVR, we can see that the maximum likelihood value increases significantly from 55,483 to 56,333 (both sets of data used).

Table 4 shows that in the EGARCH(1,1) model the variance risk premium $\lambda_2$ is negative as −0.0567 (both sets of data used) and −0.0483 (VIX data used), both significantly different than zero. The persistence of conditional variance, $\beta_1$ increases slightly from 0.9792 (returns data used) to 0.9906 (both sets of data used) and 0.9895 (VIX data used). Comparing the maximum likelihood result of the EGARCH(1,1) model under the mLRNVR and the results under the LRNVR, we can see that the maximum likelihood value increases significantly from 56,399 to 57,105 (both sets of data used) and from 33,774 to 34,303 (VIX data used).

From the comparisons in the GARCH, TGARCH, AGARCH and EGARCH models, we see that the maximum likelihood results under the mLRNVR are generally better than those under the LRNVR. Table 5 measures how the implied VIX fits the CBOE VIX by computing a list of statistics and the results demonstrate that the implied VIX under mLRNVR fits the CBOE quite well.

**Table 1.** Maximum likelihood estimates of the GARCH(1,1) model using returns data only, VIX data only or both returns and Volatility Index (VIX) data. The bold values indicate the log-likelihood value which is being maximized. The standard errors are provided in parentheses. We use a sample of 6928 daily data for the model estimation.

| | Return Only | VIX Only | | | Return & VIX | | |
|---|---|---|---|---|---|---|---|
| | | LRNVR | | mLRNVR | LRNVR | | mLRNVR |
| | | Duan (1995) | KLY (2014) | | Duan (1995) | KLY (2014) | |
| $\alpha_0$ | $2.47 \times 10^{-6}$ $(0.09 \times 10^{-6})$ | $1.68 \times 10^{-6}$ $(0.04 \times 10^{-6})$ | $1.70 \times 10^{-6}$ $(0.03 \times 10^{-6})$ | $1.64 \times 10^{-6}$ $(0.03 \times 10^{-6})$ | $1.70 \times 10^{-6}$ $(0.03 \times 10^{-6})$ | $1.68 \times 10^{-6}$ $(0.03 \times 10^{-6})$ | $1.68 \times 10^{-6}$ $(0.03 \times 10^{-6})$ |
| $\alpha_1$ | 0.1256 (0.0045) | 0.0382 (0.0006) | 0.0386 (0.0014) | 0.0456 (0.0015) | 0.0498 (0.0007) | 0.0503 (0.0007) | 0.0474 (0.0007) |
| $\beta_1$ | 0.8543 (0.0028) | 0.9351 (0.0010) | 0.9353 (0.0016) | 0.9286 (0.0036) | 0.9450 (0.0007) | 0.9467 (0.0007) | 0.9251 (0.0012) |
| $\lambda_1$ | 0.0886 (0.0116) | 0.8144 (0.0171) | 0.7980 (0.0531) | 0.2253 (0.0122) | 0.2976 (0.0085) | 0.2135 (0.0101) | 0.2134 (0.0085) |
| $\lambda_2$ | 0 | 0 | 0 | −0.3514 (0.0705) | 0 | 0 | −0.3670 (0.0195) |
| $\rho$ | 0 | 0 | −0.0039 (0.0064) | 0 | 0 | −0.0026 (0.0030) | 0 |
| $\ln L_R$ | **22,720** | 20,355 | 20,416 | 20,593 | 21,796 | 22,021 | 22,597 |
| $\ln L_V$ | 29,221 | **33,424** | **33,563** | **33,662** | 32,901 | 33,048 | 33,324 |
| $\ln L_T$ | 51,941 | 53,779 | 53,979 | 54,255 | **54,697** | **55,069** | **55,921** |

**Table 2.** Maximum likelihood estimates of TGARCH(1,1) model using returns data only, VIX data only or both returns and VIX data. The bold values indicate the log-likelihood value which is being maximized. The standard errors are provided in parentheses. We use a sample of 6928 daily data for the model estimation.

| | Return Only | VIX Only | | | Return & VIX | | |
|---|---|---|---|---|---|---|---|
| | | LRNVR | | mLRNVR | LRNVR | | mLRNVR |
| | | Duan (1995) | KLY (2014) | | Duan (1995) | KLY (2014) | |
| $\alpha_0$ | $3.45 \times 10^{-6}$ $(0.09 \times 10^{-6})$ | $1.48 \times 10^{-6}$ $(0.08 \times 10^{-6})$ | $1.48 \times 10^{-6}$ $(0.05 \times 10^{-6})$ | $1.52 \times 10^{-6}$ $(0.02 \times 10^{-6})$ | $1.48 \times 10^{-6}$ $(0.02 \times 10^{-6})$ | $1.48 \times 10^{-6}$ $(0.02 \times 10^{-6})$ | $1.48 \times 10^{-6}$ $(0.02 \times 10^{-6})$ |
| $\alpha_1$ | 0.0256 (0.0063) | 0.0020 (0.0015) | 0.0021 (0.0015) | 0.0060 (0.0010) | 0.0021 (0.0012) | 0.0021 (0.0014) | 0.0091 (0.0014) |
| $\beta_1$ | 0.8338 (0.0031) | 0.9597 (0.0007) | 0.9598 (0.0009) | 0.9553 (0.0006) | 0.9582 (0.0006) | 0.9582 (0.0006) | 0.9561 (0.0008) |
| $\theta$ | 0.1336 (0.0154) | 0.0595 (0.0026) | 0.0595 (0.0033) | 0.0531 (0.0008) | 0.0636 (0.0020) | 0.0636 (0.0021) | 0.0553 (0.0022) |
| $\lambda_1$ | 0.0131 (0.0118) | 0.3094 (0.0155) | 0.1729 (0.0317) | 0.0889 (0.0087) | 0.1729 (0.0135) | 0.1156 (0.0108) | 0.1160 (0.0075) |
| $\lambda_2$ | 0 | 0 | 0 | −0.3978 (0.0980) | 0 | 0 | −0.4112 (0.1376) |
| $\rho$ | 0 | 0 | −0.0046 (0.0060) | 0 | 0 | −0.0030 (0.0040) | 0 |
| $\ln L_R$ | **22,817** | 21,941 | 22,014 | 22,242 | 22,324 | 22,346 | 22,578 |
| $\ln L_V$ | 30,310 | **33,468** | **33,523** | **33,795** | 33,131 | 33,347 | 33,704 |
| $\ln L_T$ | 53,127 | 55,409 | 55,537 | 56,037 | **55,455** | **55,693** | **56,282** |

**Table 3.** Maximum likelihood estimates of AGARCH(1,1) model using returns data only or both returns and VIX data. The bold values indicate the log-likelihood value which is being maximized. The standard errors are provided in parentheses. We use a sample of 6928 daily data for the model estimation.

| | Return Only | Return & VIX | | |
|---|---|---|---|---|
| | | LRNVR | | mLRNVR |
| | | Duan (1995) | KLY (2014) | |
| $\alpha_0$ | $1.86 \times 10^{-6}$ $(0.08 \times 10^{-6})$ | $1.71 \times 10^{-6}$ $(0.09 \times 10^{-6})$ | $1.70 \times 10^{-6}$ $(0.05 \times 10^{-6})$ | $1.57 \times 10^{-6}$ $(0.08 \times 10^{-6})$ |
| $\alpha_1$ | 0.0841 (0.0032) | 0.0415 (0.0011) | 0.0416 (0.0013) | 0.0380 (0.0005) |
| $\beta_1$ | 0.8810 (0.0108) | 0.9302 (0.0010) | 0.9302 (0.0014) | 0.9316 (0.0012) |
| $\theta$ | 0.7861 (0.0534) | 0.7795 (0.0204) | 0.7995 (0.0456) | 0.8012 (0.0174) |
| $\lambda_1$ | 0.0255 (0.0119) | 0.0120 (0.0095) | 0.0095 (0.0120) | 0.1158 (0.0538) |
| $\lambda_2$ | 0 | 0 | 0 | −0.3125 (0.0297) |
| $\rho$ | 0 | 0 | −0.0010 (0.0058) | 0 |
| $\ln L_R$ | **22,875** | 22,298 | 22,387 | 22,690 |
| $\ln L_V$ | 29,940 | 33,185 | 33,372 | 33,643 |
| $\ln L_T$ | 52,815 | **55,483** | **55,759** | **56,333** |

**Table 4.** Maximum likelihood estimates of EGARCH(1,1) model using returns data only, VIX data only or both returns and VIX data. The bold values indicate the log-likelihood value which is being maximized. The standard errors are provided in parentheses. We use a sample of 6928 daily data for the model estimation.

| | Return Only | VIX Only | | | Return & VIX | | |
|---|---|---|---|---|---|---|---|
| | | LRNVR | | mLRNVR | LRNVR | | mLRNVR |
| | | Duan (1995) | KLY (2014) | | Duan (1995) | KLY (2014) | |
| $\alpha_0$ | −0.1919 (0.0133) | −0.0742 (0.0010) | −0.0795 (0.0015) | −0.0795 (0.0011) | −0.0845 (0.0012) | −0.0853 (0.0015) | −0.0840 (0.0012) |
| $\alpha_1$ | −0.1159 (0.0068) | −0.0622 (0.0012) | −0.0643 (0.0020) | −0.0638 (0.0012) | −0.0598 (0.0016) | −0.0650 (0.0015) | −0.0575 (0.0017) |
| $\beta_1$ | 0.9792 (0.0014) | 0.9897 (0.0002) | 0.9894 (0.0002) | 0.9895 (0.0002) | 0.9891 (0.0010) | 0.9890 (0.0003) | 0.9906 (0.0002) |
| $\kappa$ | 0.1239 (0.0081) | 0.0869 (0.0013) | 0.0886 (0.0013) | 0.0879 (0.0012) | 0.0795 (0.0021) | 0.0946 (0.0014) | 0.0817 (0.0013) |
| $\lambda_1$ | 0.0189 (0.0106) | 0.0378 (0.0201) | 0.0048 (0.0068) | 0.0294 (0.0114) | 0.0289 (0.0128) | 0.0052 (0.0043) | 0.0108 (0.0041) |
| $\lambda_2$ | 0 | 0 | 0 | −0.0483 (0.0083) | 0 | 0 | −0.0567 (0.0044) |
| $\rho$ | 0 | 0 | −0.0003 (0.0157) | 0 | 0 | −0.0022 (0.0131) | 0 |
| $\ln L_R$ | **22,862** | 22,472 | 22,506 | 22,642 | 22,645 | 22,654 | 22,813 |
| $\ln L_V$ | 29,304 | **33,774** | **33,997** | **34,303** | 33,754 | 33,928 | 34,292 |
| $\ln L_T$ | 52,166 | 56,246 | 56,503 | 56,945 | **56,399** | **56,582** | **57,105** |

**Table 5.** The table displays the related statistics between the model implied VIX and the CBOE VIX for the GARCH models during the period from 2 January 1990 to 30 June 2017. We use the sample of 6928 daily data for the model estimation and comparison. The error is computed as the difference between CBOE VIX and the implied VIX. The mean error (ME) represents the mean daily difference between the implied VIX and the CBOE VIX. The standard error (Std.Err.) represents the standard deviation of the error. The mean absolute error (MAE) calculates the mean daily absolute difference between the implied VIX and the CBOE VIX. The mean squared error (MSE) computes the mean daily squared difference between the implied VIX and the CBOE VIX. The root mean squared error (RMSE) computes the square root of the mean squared error. The *P*-value is for the null hypothesis that the implied VIX and the CBOE VIX have the same mean values. Violation of one-sigma band stands for the probability that the implied VIX lies out of the one-standard-deviation band of the CBOE VIX. The correlation coefficient (Corr. Coef.) computes the linear correlation between the implied VIX and the CBOE VIX. Autocorrelation coefficients with 1, 10, and 30 days lag and higher moments of implied VIX are also reported. Note that the calibration results using returns data only are produced by using GARCH models under the local risk-neutral valuation relationship (LRNVR). The results using VIX and both sets of data are produced by using GARCH models with the modified local risk-neutral valuation relationship (mLRNVR).

| Model&Data | ME | Std.Err. | MAE | MSE | RMSE | *P*-Value | Violation of One-Sigma Band | Corr.Coef. | AR1 | AR10 | AR30 | Variance | Skewness | Kurtosis |
|---|---|---|---|---|---|---|---|---|---|---|---|---|---|---|
| GARCH | | | | | | | | | | | | | | |
| Returns | 2.76 | 4.03 | 2.69 | 16.86 | 4.11 | 0.0000 | 7.56% | 0.90 | 0.9892 | 0.8879 | 0.6856 | 77.39 | 3.36 | 20.37 |
| VIX | 0.08 | 2.99 | 2.29 | 8.96 | 2.99 | 0.5223 | 1.68% | 0.93 | 0.9956 | 0.9452 | 0.7945 | 56.04 | 2.61 | 15.01 |
| Both | 0.16 | 3.14 | 2.25 | 9.03 | 3.01 | 0.1012 | 2.27% | 0.92 | 0.9950 | 0.9434 | 0.7853 | 56.65 | 2.69 | 15.70 |
| TGARCH | | | | | | | | | | | | | | |
| Returns | 2.44 | 4.02 | 3.40 | 22.13 | 4.70 | 0.0000 | 9.47% | 0.86 | 0.9596 | 0.7433 | 0.5327 | 53.21 | 3.33 | 21.08 |
| VIX | 0.08 | 2.96 | 2.27 | 8.80 | 2.96 | 0.5574 | 1.59% | 0.93 | 0.9955 | 0.9473 | 0.8072 | 55.88 | 2.81 | 16.83 |
| Both | 0.22 | 3.00 | 2.26 | 9.03 | 3.01 | 0.1012 | 1.80% | 0.93 | 0.9954 | 0.9431 | 0.7940 | 57.53 | 2.83 | 17.21 |
| AGARCH | | | | | | | | | | | | | | |
| Returns | 3.08 | 3.32 | 3.47 | 20.49 | 4.53 | 0.0000 | 8.14% | 0.91 | 0.9815 | 0.8495 | 0.6220 | 57.60 | 3.44 | 22.50 |
| Both | 0.19 | 2.98 | 2.27 | 8.93 | 2.99 | 0.1440 | 1.13% | 0.93 | 0.9948 | 0.9383 | 0.7774 | 57.60 | 2.92 | 17.52 |
| EGARCH | | | | | | | | | | | | | | |
| Returns | 3.39 | 3.38 | 3.62 | 22.96 | 4.79 | 0.0000 | 10.02% | 0.92 | 0.9817 | 0.8498 | 0.6408 | 35.52 | 2.08 | 10.54 |
| VIX | 0.01 | 2.71 | 2.10 | 7.38 | 2.72 | 0.8865 | 0.79% | 0.94 | 0.9946 | 0.9395 | 0.7955 | 54.37 | 2.20 | 11.12 |
| Both | 0.01 | 2.72 | 2.10 | 7.41 | 2.72 | 0.7396 | 0.82% | 0.94 | 0.9944 | 0.9373 | 0.7903 | 54.62 | 2.15 | 10.81 |
| CBOE VIX | | | | | | | | | 0.9812 | 0.8988 | 0.7580 | 61.68 | 2.10 | 10.70 |

After obtaining the estimates of the parameters in the models, we can then calculate the conditional variance $h_t$ and compute the corresponding GARCH implied VIX. Figure 1 shows the time series of the CBOE VIX and the implied VIX of the four GARCH models estimated using returns only. Figure 2 shows the time series of the CBOE VIX and the implied VIX of the GARCH(1,1) model estimated using VIX data only. Figure 3 shows the time series of the CBOE VIX and the implied VIX of the GARCH(1,1) model estimated using both returns and VIX. Similar comparison plots are obtained for other GARCH models. Specifically, the time series of the CBOE VIX and the implied VIX of the TGARCH(1,1) model estimated with VIX data only are displayed in Figure 4. The time series of the CBOE VIX and the implied VIX of the TGARCH(1,1) and AGARCH (1,1) model estimated with both returns and VIX data are shown in Figures 5 and 6, respectively. For the EGARCH(1,1) model, Figure 7 shows the comparison between the CBOE VIX and model implied VIX with VIX data only, and Figure 8 displays the comparison between the CBOE VIX and model implied VIX with both returns and VIX data. From the list of graphs, we observe that the model implied VIX fits the CBOE VIX better under the mLRNVR compared with the LRNVR in general. In particular, the ratios of the implied VIX values to the CBOE VIX values are closer to 1 under the mLRNVR compared to those under the LRNVR, as shown in Figures 3, 5, 6 and 8. The direct comparisons of performance under LRNVR and mLRNVR for the GARCH, TGARCH, AGARCH and EGARCH models are listed in Tables 1–4, respectively.

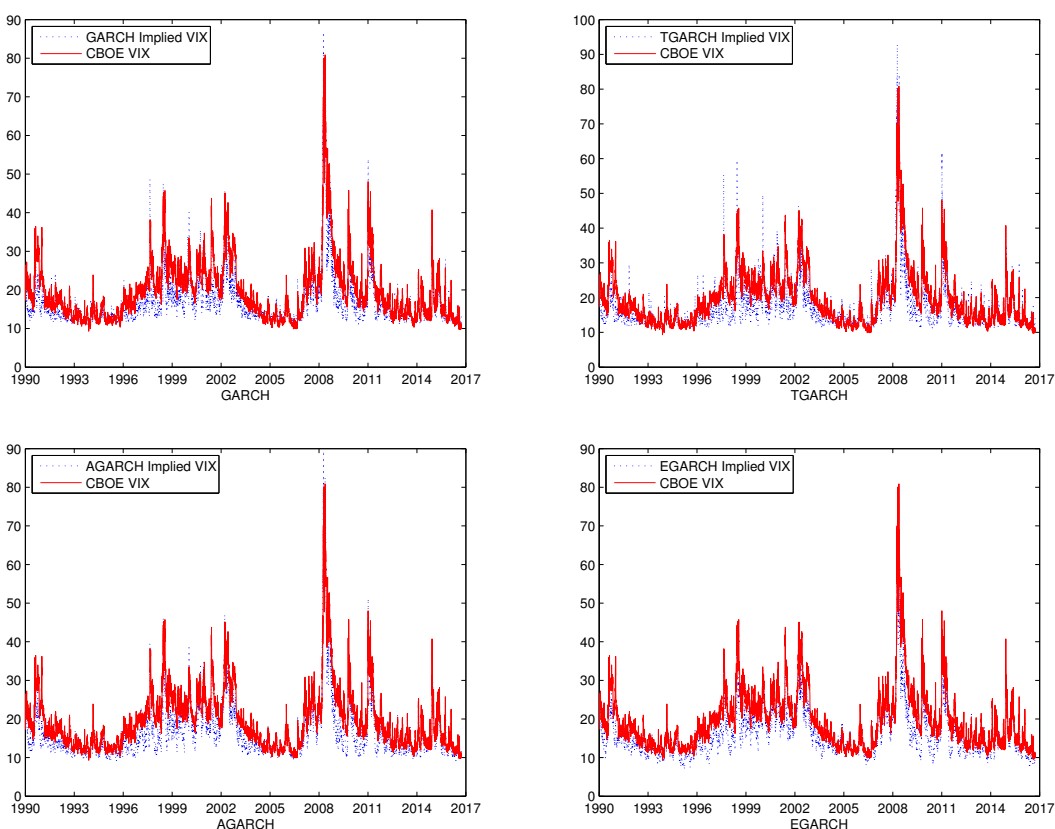

**Figure 1.** Comparison between CBOE VIX and implied VIX using returns data only for four GARCH models under the LRNVR.

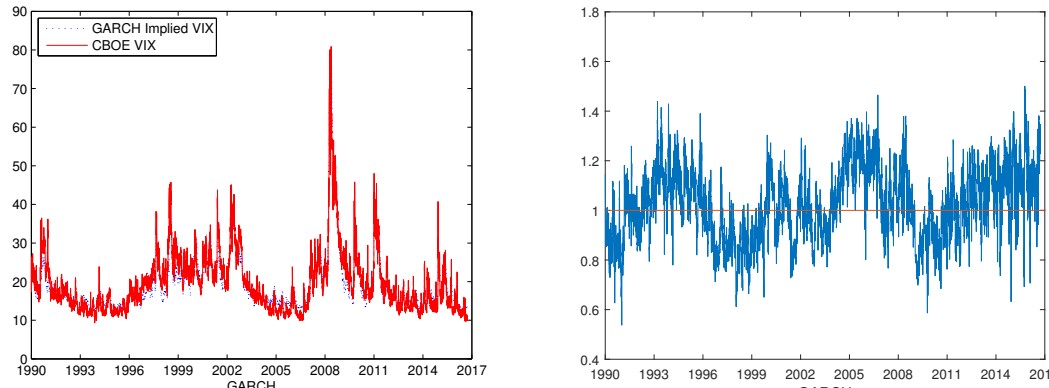

**Figure 2.** Comparison between CBOE VIX and implied VIX of the GARCH(1,1) model using VIX data only. The left panel shows the index values of CBOE VIX and implied VIX and the right panel shows the ratio of the implied VIX to CBOE VIX and a horizontal line at 1 for reference.

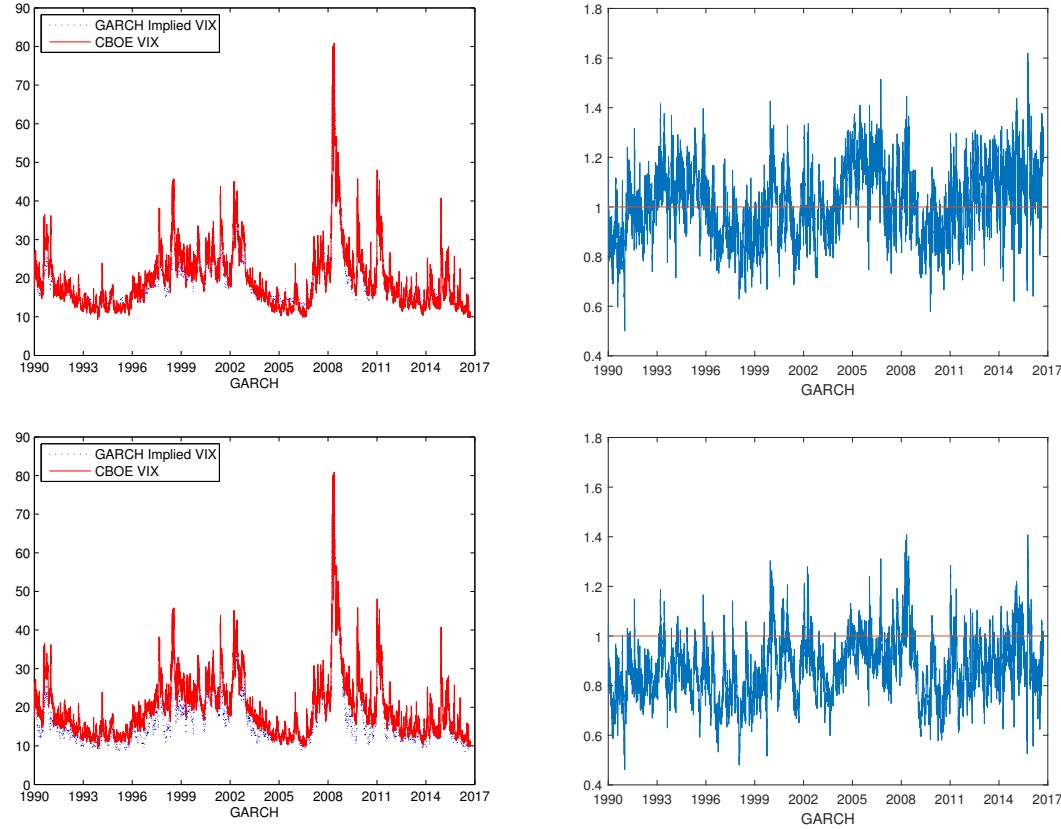

**Figure 3.** Comparison between CBOE VIX and implied VIX of the GARCH(1,1) model using both returns and VIX data with the upper panels showing the result under the mLRNVR and the lower panels showing the result under the LRNVR. The left panels show the index values of CBOE VIX and implied VIX and the right panels show the ratio of the implied VIX to CBOE VIX and a horizontal line at 1 for reference.

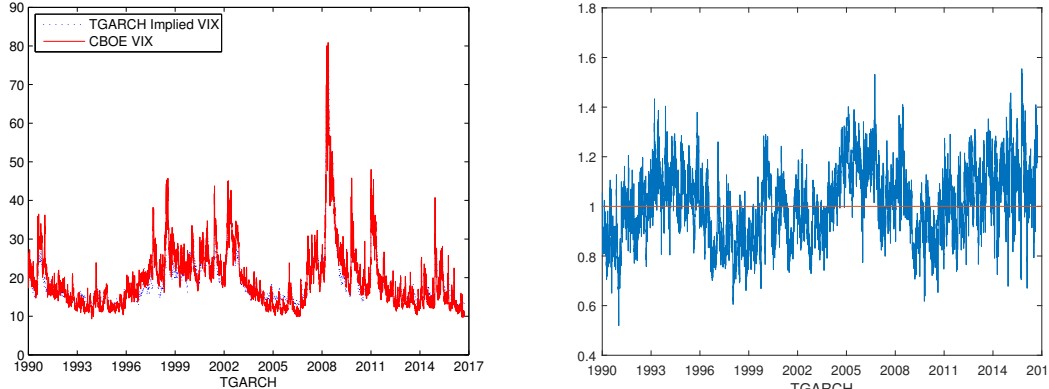

**Figure 4.** Comparison between CBOE VIX and implied VIX of the TGARCH(1,1) model using VIX data only. The left panel shows the index values of CBOE VIX and implied VIX and the right panel shows the ratio of the implied VIX to CBOE VIX and a horizontal line at 1 for reference.

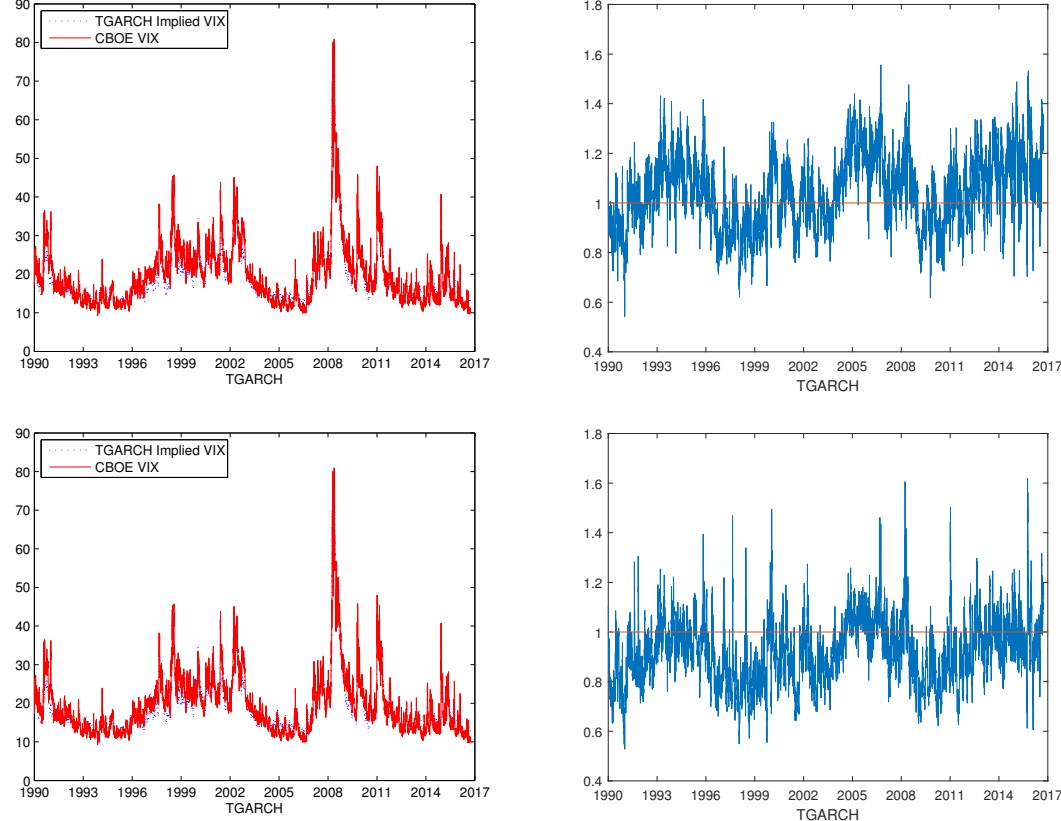

**Figure 5.** Comparison between CBOE VIX and implied VIX of the TGARCH(1,1) model using both returns and VIX data with the upper panel showing the result under the mLRNVR and the lower panel showing the result under the LRNVR. The left panels show the index values of CBOE VIX and implied VIX and the right panels show the ratio of the implied VIX to CBOE VIX and a horizontal line at 1 for reference.

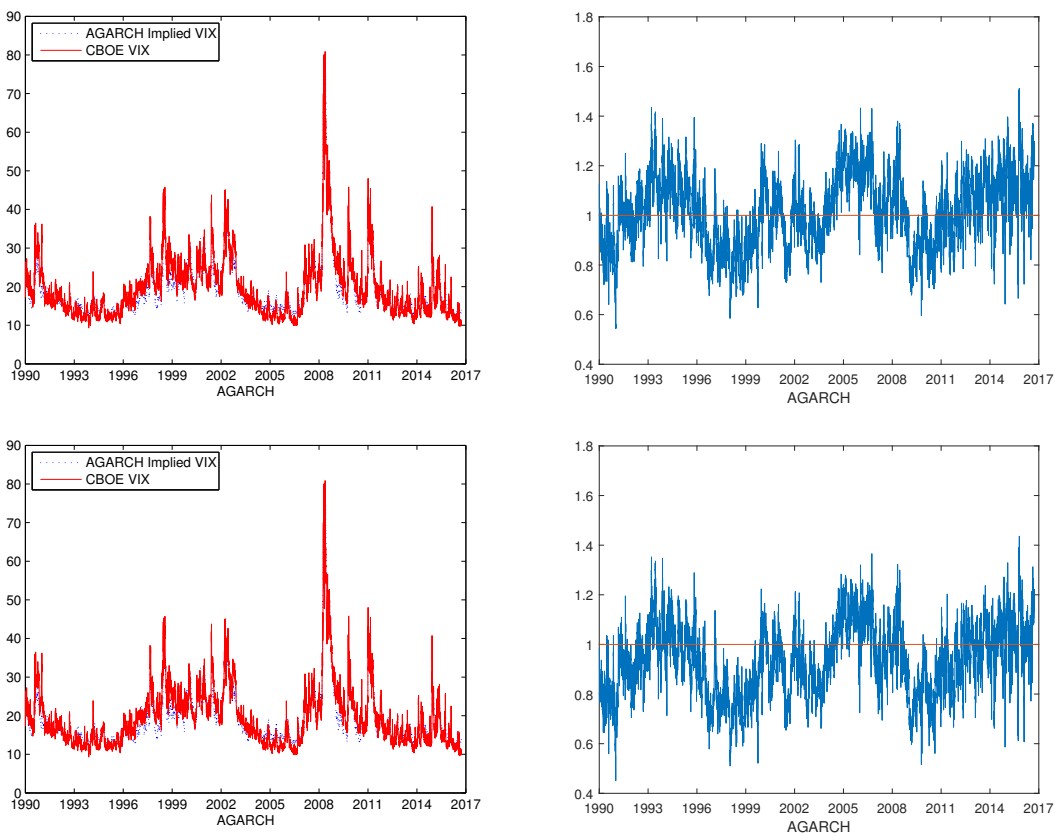

**Figure 6.** Comparison between CBOE VIX and implied VIX of the AGARCH(1,1) model using both returns and VIX data with the upper panel showing the result under the mLRNVR and the lower panel showing the result under the LRNVR. The left panels show the index values of CBOE VIX and implied VIX and the right panels show the ratio of the implied VIX to CBOE VIX and a horizontal line at 1 for reference.

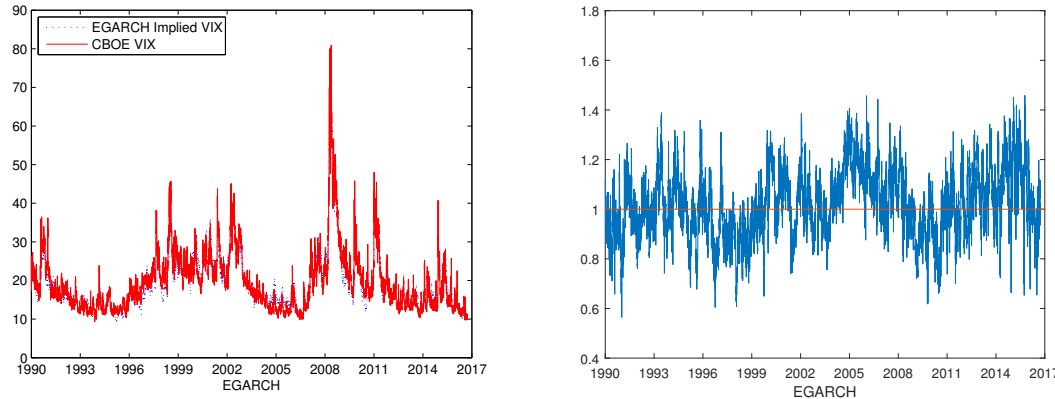

**Figure 7.** Comparison between CBOE VIX and implied VIX of the EGARCH(1,1) model using VIX data only. The left panel shows the index values of CBOE VIX and implied VIX and the right panel shows the ratio of the implied VIX to CBOE VIX and a horizontal line at 1 for reference.

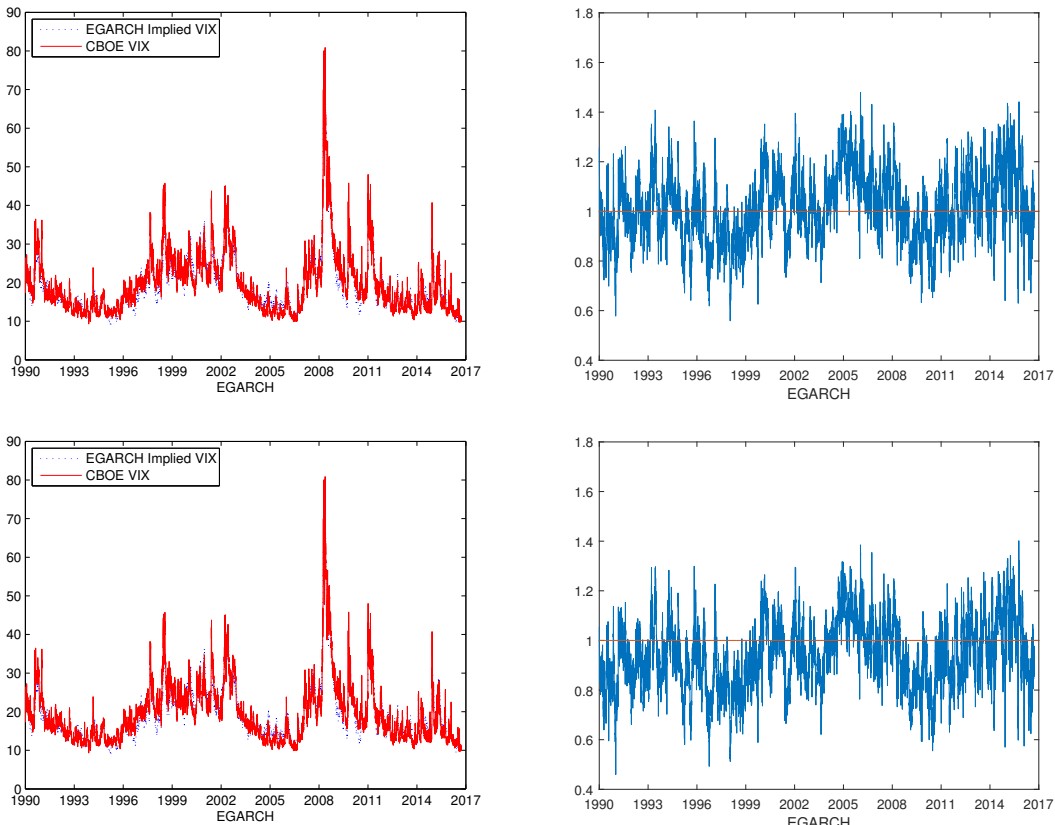

**Figure 8.** Comparison between CBOE VIX and implied VIX of the EGARCH(1,1) model using both returns and VIX data with the upper panel showing the result under the mLRNVR and the lower panel showing the result under the LRNVR. The left panels show the index values of CBOE VIX and implied VIX and the right panels show the ratio of the implied VIX to CBOE VIX and a horizontal line at 1 for reference.

## 6. Theoretical Justification

Duan studied the bivariate diffusion limit of the GARCH(1,1) model as the length of the time period tends towards zero in Duan (1996, 1997). Applying Duan's arguments, one can show that the limiting bivariate diffusion process of the approximating GARCH$(1,1)$ process under the physical measure $P$ is given by

$$
\begin{aligned}
d \ln X_t &= \left( r - \frac{1}{2} h_t + \lambda_1 \sqrt{h_t} \right) dt + \sqrt{h_t} dW_{1t}, \\
dh_t &= (\alpha_0 + (\alpha_1 + \beta_1 - 1) h_t) dt + \sqrt{2} \alpha_1 h_t dW_{2t}, \\
&= (\alpha_0 + (\alpha_1 + \beta_1^* - 1) h_t) dt + \sqrt{2} \alpha_1 \lambda_2 h_t dt + \sqrt{2} \alpha_1 h_t dW_{2t},
\end{aligned}
\tag{27}
$$

where the persistence parameter of conditional variance is defined as $\beta_1^* = \beta_1 - \sqrt{2} \alpha_1 \lambda_2$ under the mLRNVR. The terms $dW_{1t}$ and $dW_{2t}$ are independent standard Brownian motions under the physical measure $P$. The limiting bivariate diffusion under the risk-neutral measure $Q$ is a re-parameterization of Hull and White (1987) bivariate diffusion model as follows:

$$
\begin{aligned}
d \ln X_t &= \left( r - \frac{1}{2} h_t \right) dt + \sqrt{h_t} dZ_{1t} \\
dh_t &= (\alpha_0 + (\alpha_1 + \beta_1^* - 1) h_t) dt + \sqrt{2} \alpha_1 h_t dZ_{2t},
\end{aligned}
\tag{28}
$$

where $dZ_{1t} = dW_{1t} + \lambda_1 dt$ and $dZ_{2t} = dW_{2t} + \lambda_2 dt$ are independent standard Brownian motions under the mLRNVR $Q$. Both equity risk premium $\lambda_1$ and variance risk premium $\lambda_2$ are present in the

model under the risk-neutral measure $Q$. The discrete-time GARCH$(1,1)$ process (5) corresponds to the limiting diffusion process under the risk-neutral measure $Q$.

## 7. Conclusions

In this paper, we follow the GARCH option-pricing framework of Duan (1995) and propose a modified local risk-neutral valuation relationship. The new risk-neutral valuation is referred to as the mLRNVR. The advantage of the mLRNVR compared with the LRNVR commonly used in the literature (Duan (1995); Hao and Zhang (2013); Wang et al. (2017)) is that the variance risk premium is included in the risk-neutral dynamics under the mLRNVR. The absence of a variance risk premium in the risk-neutral dynamics under the LRNVR is noted in Hao and Zhang (2013), where it is shown that both empirical studies and theoretical results indicated that the GARCH models under the LRNVR did not capture the variance premium.

We then find the theoretical VIX squared value as the conditional risk-neutral expectation of the arithmetic mean variance over the next 21 trading days under the mLRNVR. Specifically, the GARCH implied VIX formulas are derived using the features of square-root stochastic autoregressive volatility (SR-SARV) models. We apply several calibration methods to estimate the model parameters using various sets of time series data, and compare the theoretical formula performances with the market data. Various combinations of the time series data of the daily closing price of the S&P 500 index and the CBOE VIX are used to find the maximum likelihood estimation of the GARCH models. The corresponding implied VIX time series are then calculated from the calibrated model. Similar to the empirical evidence in Hao and Zhang (2013) and Wang et al. (2017), when only the S&P 500 returns are used for estimation, the GARCH implied VIX is consistently and significantly lower than the CBOE VIX. When the CBOE VIX is used for estimation, the implied VIX fits the statistical properties of the CBOE VIX and matches the CBOE VIX data better. The numerical results provide evidence that the GARCH option pricing under the mLRNVR is more suitable to price volatility. In particular, the numerical results show that the EGARCH model using both returns and VIX data under the mLRNVR provides the best fit to the sample data. Therefore, we recommend that the option pricing in the GARCH framework should use the EGARCH model with returns and VIX data under the mLRNVR for the best results in the future. In the case of GARCH(1,1), we also compare the diffusion limit of the GARCH process under the physical measure and the mLRNVR risk-neutral measure to show that the variance premium is captured in the risk-neutral dynamics.

**Author Contributions:** Conceptualization, J.E.Z.; methodology, J.E.Z.; software, W.Z.; validation, J.E.Z. and W.Z.; formal analysis, W.Z.; investigation, W.Z. and J.E.Z.; resources, J.E.Z. and W.Z.; data curation, W.Z.; writing–original draft preparation, W.Z.; writing-review and editing, J.E.Z.; visualization, W.Z.; supervision, J.E.Z.; project administration, W.Z.; funding acquisition, J.E.Z. All authors have read and agreed to the published version of the manuscript.

**Funding:** Jin E. Zhang has been supported by an establishment grant from the University of Otago and the National Natural Science Foundation of China grant (Project No. 71771199).

**Acknowledgments:** We are grateful to Xinfeng Ruan, Yanlin Shi (our AFM discussant) and the seminar participants at the 2017 Auckland Financial Meeting (AFM) in Queenstown for helpful comments and suggestions. Wenjun Zhang acknowledges the financial support from the Auckland University of Technology and the hospitality from the University of Otago.

**Conflicts of Interest:** We declare that we have no relevant or material financial interests that relate to the research described in this paper.

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
