# Peer review of "GARCH Option Pricing Models and the Variance Risk Premium"

_jrfm, doi:10.3390/jrfm13030051_

Round 1
Reviewer 1 Report
The paper following the GARCH option-pricing framework of Duan (1995) proposes a modified local risk-neutral valuation relationship (mLRNVR). The proposed mLRNVR compared with the Duan’s LRNVR includes the variance risk premium. Comparison of estimation results for LRNVR and mLRNVR show that mLRNVR process has larger maximum likelihood values.
The paper has some points which need improvement and further explanation:
It is not explained why Dian and other researchers who have followed him/her have not included variance risk premium; The Duan’s equation for return excludes return risk premium, but Duan’s variance equation includes the risk premium of the return. The risk premium of the return is explained by the standard deviation (the square root of the variance). Is it important to included the premium for the variance again? Why? The risk premium for variance included is not explained why it takes such a form. Why the persistence coefficient takes such a form in equation 3. It is proposed but not explained. The theoretical justification does not justify the inclusion of the proposed variance risk premium in Section 6. The likelihood values for mLRNVR are larger, but the figures showing the derived VIX with mLRNVR and LRNVR look similar in Figs. 3, 5 and 6. The is not any explanation of these data in the paper.
Note: I have not checked the paper for plagiarism.
Reviewer 2 Report
The paper proposes a modification to Duan’s (1995) risk-neutral valuation relationship for GARCH-type option pricing models. The modification is labelled mLRNVR. The paper builds on the contribution of Hao and Zhang (2013) who found that GARCH implied VIX is unable to predict the CBOE VIX, when the models are estimated under LRNVR. They identify the problem being GARCH models under LRNVR failing to incorporate the variance risk premium.
The impact of the variance risk premium is that the variance process is more persistent in the risk-neutral measure than in the physical one. This seems to be borne out by the significantly negative estimate of the parameter lambda_2 in the econometric results. Perhaps this should be stressed; a negative value of lambda_2 is expected because the term involving lambda_2 in (3) has a negative sign.
Daily data on the S&P500 used to demonstrate that GARCH-type models estimated under mLRNVR are able to predict the CBOE VIX accurately. On this basis, I would say that the paper is making a useful contribution. However, there are various ways in which the exposition could be improved.
I think it is confusing that the name “VIX” is being used to represent several different things. Firstly, it is used to represent the CBOE VIX (computed using option prices) as described in the first paragraph of Section 3. Secondly, it is defined as a mathematical expectation in an unnumbered equation on p.11. Thirdly, it is defined as a sample mean over trading days in an un-numbered equation on p.12. I note that later in the paper the superscripts “mkt” and “imp” are used. My recommendation is that this sort of notation is used at the time of defining the the various quantities.
The principal method of model comparison is a sequence of time series graphs, each comparing the model prediction of VIX to the CBOE VIX. I must say that it is quite hard to compare the models on the basis of these plots. I would recommend plotting the CBOE VIX just once, and in the other graphs present the ratio of the prediction to the CBOE VIX, ensuring that all the graphs are drawn using the same vertical scale. I think this would make it much easier to compare the performance of the various models. It might even be possible to present groups of ratios on the same graph.
It seems that the models are compared on the basis of in-sample performance only. I assume practitioners are more interested in out-of-sample performance. Is it possible to assess out-of-sample predictive performace of these models?
Minor points
Shouldn’t the title be “Garch Option Pricing Models and the Variance Risk Premium”?
Actually, I think the title of the paper is slightly misleading because the paper is only indirectly related to option pricing.
Abstract: “GARCH…under mLRNVR..….able to price VIX correctly”. “correctly” seems too strong a claim. Do you mean “accurately”?
Why are only some equations numbered?
p.8 After equation (2), you refer to the “asset returns process in (1)”, but (1) does not appear to be an asset returns process. The Asset returns process is specified in the un-numbered equation on p.7.
p.9 Bottom. “persistence parameter beta_1 is …different in the P and Q measures”. This is careless because in the Q measure the persistence parameter is not beta_1, it is beta_1-root2 alpha_1 lambda_2.
p.10. 3 lines from end. “We adopt the idea by….”. “adopt” or “adapt”?
p.17. End of first paragraph of Section 4. “VIX times series” should be “VIX time series”.
p.18. In equation (13), X_t is not defined.
p.30. In all tables of results, the sample size should be included.
Round 2
Reviewer 1 Report
The revised version is enough improved and suitable for publication.
Author Response
Thank you for accepting the paper.
Reviewer 2 Report
Thanks for responding to the points made in the previous report. Most of the points have been adequately addressed. Some issues remain:
- p.10. The newly inserted text contains the term "risk-neural". This should be "risk-neutral".
- The new graphs in Figs 2-7 look good. However, they will be much easier to interpret if they contain a horizontal line at one. (If you're using STATA, use the option yline(1)). Then it becomes very obvious that the LRNVR tends to produce downward-biased predictions while the mLRNVR reduced this bias.
- In both Figures 6 and 8, the bottom graphs appear identical to the top graphs. Also, Figure 7 appears identical to Figure 8. I think errors have been made in obtaining these graphs.
Author Response
Response to Reviewer 2 Comments
Point 1: p.10. The newly inserted text contains the term "risk-neural". This should be "risk-neutral".
Response 1:
Thank you for your comment. We have changed ‘risk-neural’ to ‘risk-neutral’.
Point 2: The new graphs in Figs 2-7 look good. However, they will be much easier to interpret if they contain a horizontal line at one. (If you're using STATA, use the option yline(1)). Then it becomes very obvious that the LRNVR tends to produce downward-biased predictions while the mLRNVR reduced this bias.
Response 2:
Thank you for your comment. We have added a horizontal line at 1 for reference in Figs 2-7.
Point 3: In both Figures 6 and 8, the bottom graphs appear identical to the top graphs. Also, Figure 7 appears identical to Figure 8. I think errors have been made in obtaining these graphs.
Response 3:
Thank you for your comment. We have checked the figures again. We agree that they appear to be very similar due to the graphs in different panels follow the same type of models with different parameters. But if we look closely, there are some slight differences between each panels.
